# MINIMALIST EXPLANATION GENERATION AND CIRCUIT DISCOVERY

## ABSTRACT

Machine learning models, by virtue of training, learn a large repertoire of decision rules for any given input, and any one of these may suffice to justify a prediction. However, in high-dimensional input spaces, such rules are difficult to identify and interpret. In this paper, we introduce an activation-matching–based approach to generate minimal and faithful explanations for the decisions of pre-trained image classifiers. We aim to identify minimal explanations that not only preserve the model's decision but are also concise and human-readable. To achieve this, we train a lightweight autoencoder to produce binary masks that learns to highlight the decision-wise critical regions of an image while discarding irrelevant background. The training objective integrates activation alignment across multiple layers, consistency at the output label, priors that encourage sparsity, and compactness, along with a robustness constraint that enforces faithfulness. The minimal explanations so generated also lead us to mechanistically interpreting the model internals. In this regard we also introduce a circuit readout procedure wherein using the explanation's forward pass and gradients, we identify active channels and construct a channel-level graph, scoring inter-layer edges by ingress weight magnitude times source activation and feature-to-class links by classifier weight magnitude times feature activation. Together, these contributions provide a practical bridge between minimal input-level explanations and a mechanistic understanding of the internal computations driving model decisions.

## 1 INTRODUCTION

The ability to generate explanations is key to making the decisions of modern machine learning models transparent and trustworthy. While deep neural networks achieve impressive predictive accuracy, their outputs arise from complex, high-dimensional computations that are not directly interpretable. Such models learn a wide range of decision rules during training, and any one of these may suffice for a given input. Without explanations, however, it is difficult to determine which rule was used or which aspects of the input were responsible for the prediction. The opacity of such processes means that the precise basis of a decision often remains hidden, limiting transparency and accountability.

A natural way to make explanations more interpretable is to focus on minimality. By isolating the smallest subset of input features sufficient to support a prediction, one obtains an explanation that is both faithful to the model's internal computation and human-readable. Minimal explanations highlight a compact subset of pixels in the case of images, or features in general, that directly support the output. Such explanations serve not only as cognitive aids for human understanding but also as a practical diagnostic tool: they can explain counterfactual behaviors, highlight shortcut learning, and reveal when the model relies on inappropriate evidence. This is critical in safety-sensitive applications such as medical diagnostics, autonomous driving, and security, where knowing the precise basis for a decision can determine whether the system is trustworthy.

In this work, we propose an *activation-matching* based approach that, given an input image and a frozen pretrained classifier, trains a lightweight autoencoder to generate a binary mask selecting the minimal set of pixels needed to preserve the model's prediction and internal activations. We further leverage these explanations to uncover concise, channel-level views of the model's computation, revealing sparse, data-dependent subcircuits sufficient for the decision. Beyond the forward activation pathways, we also couple this analysis with gradient information, tracing back the most

prominent gradients from the output class logit toward earlier layers and the input. This dual perspective—forward activations and backward gradients—enables us to connect minimal input-level explanations with both mechanistic insight into signal flow and attribution of decision-critical pathways, yielding a more comprehensive understanding of how deep networks arrive at their predictions.

## 2 PRIOR WORK

### 2.1 MODEL EXPLAINABILITY VIA INVERSION

Inversion techniques aim to reconstruct input patterns that trigger specific outputs or internal activations in a neural network. Unlike explanations, which are inherently tied to a particular input and its corresponding decision, inversion seeks to synthesize representative stimuli that reveal what a model has learned in aggregate. Early work on multilayer perceptrons applied gradient-based inversion to visualize decision boundaries, though the resulting reconstructions were often noisy or adversarial-like Kindermann & Linden (1990); Jensen et al. (1999); Saad & Wunsch (2007). To address these limitations, researchers explored evolutionary search and constrained optimization Wong (2017). Subsequent advances incorporated prior-based regularization, such as smoothness constraints or pretrained generative models, which enhanced both realism and interpretability of reconstructions Mahendran & Vedaldi (2014); Yosinski et al. (2015); Mordvintsev et al. (2015); Nguyen et al. (2016; 2017). More recently, methods have emerged that stabilize inversion by learning surrogate loss landscapes Liu et al. (2022), while generative approaches conditionally reconstruct inputs likely to produce a given output Suhail & Sethi (2024). Alternative formulations as in Suhail (2024) encode classifiers into CNF constraints, framing inversion as a deterministic satisfiability problem.

### 2.2 INPUT-LEVEL EXPLAINABILITY

While inversion focuses on global characterizations of model behavior, input-level explanation methods aim to provide faithful rationales for specific predictions. Explainable AI has thus developed into a central research field Ali et al. (2023); Hsieh et al. (2024); Gilpin et al. (2018), driven by the demand for trust, transparency, and accountability in high-stakes applications. Among post-hoc attribution methods, LIME builds local surrogate models to approximate decision boundaries Hamilton et al. (2022), whereas Grad-CAM highlights salient image regions through gradient-weighted activations Selvaraju et al. (2019). More recent advances emphasize concept-based explanations that map predictions to semantically interpretable parts Lee et al. (2025). Evaluating such methods remains an open challenge: surveys have highlighted the importance of rigorous metrics combining fidelity, stability, and human-centered evaluation Zhou et al. (2021). Explanations are also being embedded into interactive systems, allowing users to guide, debug, or refine models through explanation-driven feedback Teso et al. (2022). Beyond heuristics, Ignatiev et al. (2018) also explore abductive reasoning approaches that provide subset- or cardinality-minimal explanations with formal guarantees.

### 2.3 MECHANISTIC INTERPRETABILITY OF CIRCUITS

Mechanistic interpretability investigates the *circuits* within a model—sparse subnetworks of neurons and connections that implement particular algorithms. Minimal explanations can reveal the smallest sufficient evidence for a prediction, offering insights into how internal components drive decisions. Early studies of circuits relied on manual inspection, limiting scalability. Recent approaches automate this process: ACDC Conmy et al. (2023) introduced a systematic framework that rediscovered known transformer circuits through activation patching. Building on this, Rajaram et al. (2024) extended circuit discovery to vision models, extracting subnetworks responsible for concept recognition and demonstrating that targeted edits could alter predictions and enhance robustness. Further work Nainani et al. (2024) explored how circuits generalize across diverse inputs, revealing that networks often reuse a shared set of components while flexibly adapting their connectivity—a manifestation of representational superposition.

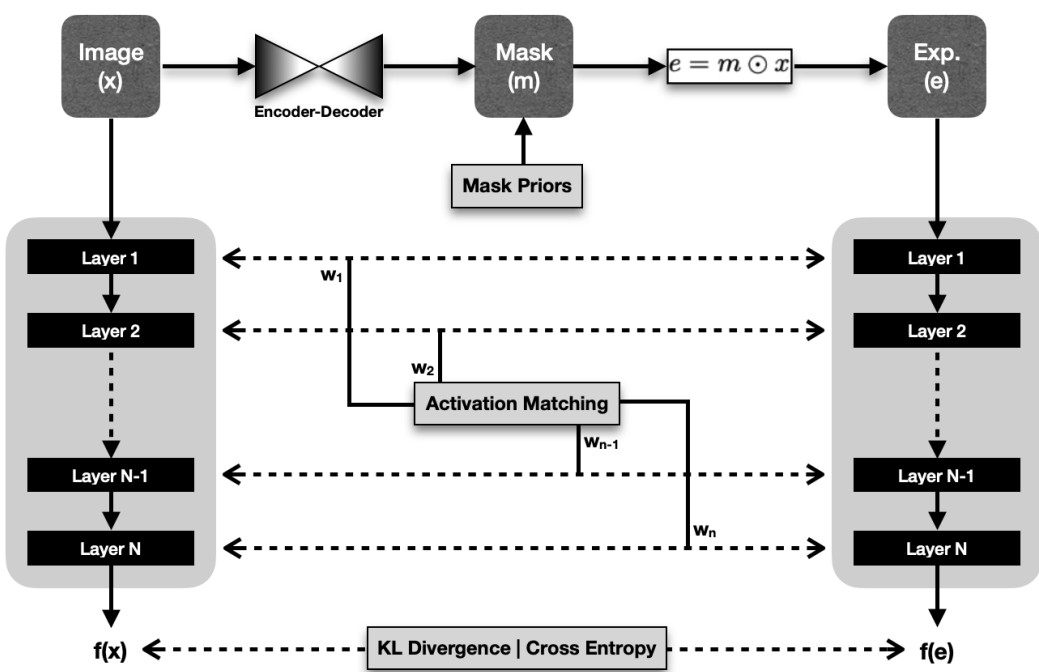

Figure 1: Schematic Approach to generating mask 'm' and explanation 'e' for an image 'x' by matching activations across different layers of a frozen classifier 'f'.

## 3 METHODOLOGY

We aim to generate minimal, faithful explanations for a frozen classifier $f$ on any input image $x$, and to use these explanations to expose compact internal circuits that drive its decisions. Our framework has two stages: (i) explanation generation via activation alignment and sparsity-inducing priors as shown in Figure 1, and (ii) circuit discovery from explanation-induced activations and gradients.

### 3.1 EXPLANATION GENERATION

In order to generate explanations we train a lightweight autoencoder to produce a binary mask $m$, defining the explanation as $e = m \odot x$, which suppresses irrelevant regions. The autoencoder is optimized with a composite objective whose terms are weighted to balance fidelity, sparsity, smoothness, and robustness.

#### 3.1.1 ACTIVATION MATCHING AND OUTPUT FIDELITY

**Weighted activation matching**

$$\mathcal{L}_{\mathrm{act}} = \sum_{\ell} \alpha_\ell \, d\big(\phi_\ell(x), \phi_\ell(e)\big)$$

This loss aligns post-ReLU features $\phi_\ell$ of $x$ and $e$ across layers, with per-layer weights $\alpha_\ell$ emphasizing deeper or shallower representations as needed. The exact form of the distance function $d(\cdot, \cdot)$ depends on the layer type: for convolutional feature maps, mean squared error (MSE) is appropriate, while for linear layers, cosine similarity provides a natural choice. Together, these ensure that the explanation follows the same internal computation trajectory as the original input.

**Cross-entropy loss**

$$\mathcal{L}_{\mathrm{CE}} = -\log p_{f(e)}(y)$$

Cross Entropy is used to preserves the top-1 label $y$ predicted from $x$, ensuring that the explanation remains decisional-equivalent to the original image. It prevents degenerate masks that match features but flip the class.

**KL divergence loss**

$$\mathcal{L}_{\mathrm{KL}} = D_{\mathrm{KL}}\big(\mathrm{softmax}(f(x)) \,\|\, \mathrm{softmax}(f(e))\big)$$

In order to match the full predictive distributions, not just the argmax, between the output for the image and the explanation, KL Divergence is used. This stabilizes training and discourages explanations that achieve correctness while distorting non-target probabilities.

### 3.1.2 MASK PRIORS FOR MINIMALITY

**Area loss**

$$\mathcal{L}_{\mathrm{area}} = \|m\|_1$$

In the interest of minimality, we directly penalize active pixels, pushing the mask toward the smallest region sufficient to preserve the decision. A higher weight yields more compact, human-readable explanations by increasing sparsity.

**Binarization loss**

$$\mathcal{L}_{\mathrm{bin}} = \|m - m^2\|_1$$

We penalize mask values that lie between 0 and 1 so the encoder learns to either include or exclude pixels entirely, driving values toward $\{0, 1\}$. This produces sharp boundaries rather than fuzzy heatmaps. To enable end-to-end training through the non-differentiable threshold, we use a straight-through estimator (STE), treating the binarization as identity in the backward pass.

**Total variation loss**

$$\mathcal{L}_{\mathrm{tv}} = \sum_{i,j} \big(|m_{i,j} - m_{i+1,j}| + |m_{i,j} - m_{i,j+1}|\big)$$

While TV is not strictly required for minimality, using the area loss alone often activates sparse, non-contiguous speckles that are less interpretable. To generate contiguous and compact masks, we pair the area loss with total variation, which suppresses isolated activations and encourages smooth, coherent regions.

### 3.1.3 ABDUCTIVE ROBUSTNESS CONSTRAINT

Given a random background $r$, we perturb the explanation by replacing the pixels outside the mask with $r$ sampled from Gaussian noise:

$$\tilde{e} = m \odot x + (1 - m) \odot r.$$

We then enforce that the classifier predicts the same label as for the original image/explanation by applying a cross-entropy penalty,

$$\mathcal{L}_{\mathrm{rob}} = -\log p_{f(\tilde{e})}(y),$$

where $y$ is the class predicted for $x$. This constraint operationalizes the notion of sufficiency that the pixels retained by the mask must contain all the evidence needed for the decision, so arbitrary perturbations to the complement should not alter the outcome. This robustness term discourages solutions that inadvertently exploit background cues or dataset-specific artifacts, and complements the minimality priors by ensuring that the explanation is not only small and crisp, but also reliable under changes outside the highlighted region.

## 3.2 TRAINING OBJECTIVE.

We train the autoencoder using $\mathcal{L}_{\mathrm{EXP}}$ a weighted sum of activation matching terms, minimality priors over the mask, and a robustness constraint. For clarity, we group the components as:

$$
\begin{aligned}
\mathcal{L}_{\mathrm{AM}} &= \lambda_{\mathrm{act}}\mathcal{L}_{\mathrm{act}} + \lambda_{\mathrm{CE}}\mathcal{L}_{\mathrm{CE}} + \lambda_{\mathrm{KL}}\mathcal{L}_{\mathrm{KL}} && \big[\text{activation matching \& output fidelity}\big] \\
\mathcal{L}_{\mathrm{MIN}} &= \lambda_{\mathrm{area}}\mathcal{L}_{\mathrm{area}} + \lambda_{\mathrm{bin}}\mathcal{L}_{\mathrm{bin}} + \lambda_{\mathrm{tv}}\mathcal{L}_{\mathrm{tv}} && \big[\text{mask priors for minimality}\big] \\
\mathcal{L}_{\mathrm{ROB}} &= \lambda_{\mathrm{rob}}\mathcal{L}_{\mathrm{rob}} && \big[\text{robustness constraints}\big]
\end{aligned}
$$

$$\mathcal{L}_{\mathrm{EXP}} = \mathcal{L}_{\mathrm{AM}} + \mathcal{L}_{\mathrm{MIN}} + \mathcal{L}_{\mathrm{ROB}}.$$

By tuning the coefficients $\{\lambda_\cdot\}$ to the task, the encoder learns binary masks that are minimal, sharp, and contiguous while remaining decisional-equivalent and robust to perturbations outside the highlighted region.

## 3.3 CIRCUIT DISCOVERY

Beyond input-level explanations, our approach provides a window into the network's internal computations. Given the explanation $e$, we pass it through the frozen classifier $f$ and collect activations at successive layers. For each convolutional block, we rank channels by their activation energy

$$E_c = \sqrt{\sum_{i,j} \phi_\ell(e)_{c,i,j}^2},$$

and retain the top-$k$ channels as nodes. This selects only the most influential feature maps, yielding a sparse representation of the computation. In addition to activations, we also collect backpropagated gradients with respect to these channels, which highlight features most responsible for the output. Combining forward activations with backward sensitivities provides a more faithful picture of salience.

Edges between layers are scored by combining structural weights and functional activations. For a destination channel $d$ in layer $\ell + 1$ and a source channel $s$ in layer $\ell$, we define

$$w_{s \to d} = \left\| W_{d,s}^{(\ell)} \right\|_1 \cdot E_s,$$

where $W_{d,s}^{(\ell)}$ is the convolutional kernel connecting $s$ to $d$, and $E_s$ is the energy of the source channel.

In parallel, gradient-based edge weights can be computed by scaling $W_{d,s}^{(\ell)}$ with the gradient magnitude at the destination, tracing how strongly perturbations at the output flow back toward earlier channels.

Finally, connections from the penultimate feature vector $h \in \mathbb{R}^{512}$ to class logits are scored by

$$w_{h_j \to y} = \left| W_{y,j}^{(\text{fc})} \right| \cdot |h_j|,$$

where $W_{y,j}^{(\text{fc})}$ is the fully connected weight to class $y$ from feature dimension $j$. Here too, we augment with gradient information, weighting by the sensitivity of the logit to each feature dimension.

The resulting graph highlights a compact *subcircuit* of nodes and edges that suffices for the prediction. Interpretability arises because these subcircuits are both data-dependent and minimal: irrelevant channels are pruned away by the mask, leaving behind only the critical flow of information. Incorporating gradients ensures that not only strong forward activations, but also the most decisive backward attributions, are represented. Such circuit visualizations reveal not only which pixels of the input matter (through the explanation mask), but also how that evidence propagates and feeds back through successive layers to drive the decision. In practice, this allows us to bridge input-level interpretability with mechanistic insight into the model's internal structure, exposing sparse computational pathways that underpin each classification.

## 4 RESULTS

Our approach is fairly general, and we evaluate it on a diverse set of pretrained classifiers spanning both standard and custom architectures. Specifically, we report results on ResNet-18, MobileNet-V3, ConvNeXt-Small, EfficientNet, ViT-16 pre-trained on ImageNet, as well as custom CNNs trained on MNIST. For each backbone, we employ a lightweight U-Net–based encoder to generate a binary explanation mask. Both the original image and its masked counterpart are passed through the frozen network, and we tap post activations outputs at multiple layers together with the final logits. These activations are aligned using mean squared error, while predictive outputs are matched through KL divergence and cross-entropy. To enforce minimality, we place strong emphasis on the area loss in conjunction with the robustness constraint, yielding crisp and faithful explanations that generalize across architectures of varying depth, parameterization, and inductive biases.

Figure 2 illustrates an example for an image from the ImageNet class *EntleBucher* passed through a pre-trained resnet-18 model. The first row shows the original image, the binary mask, and the resulting explanation. The second row compares the circuit graphs obtained from the original image and from the explanation when passed through the ResNet. The forward pass is represented by black while the gradients are represented by red lines.

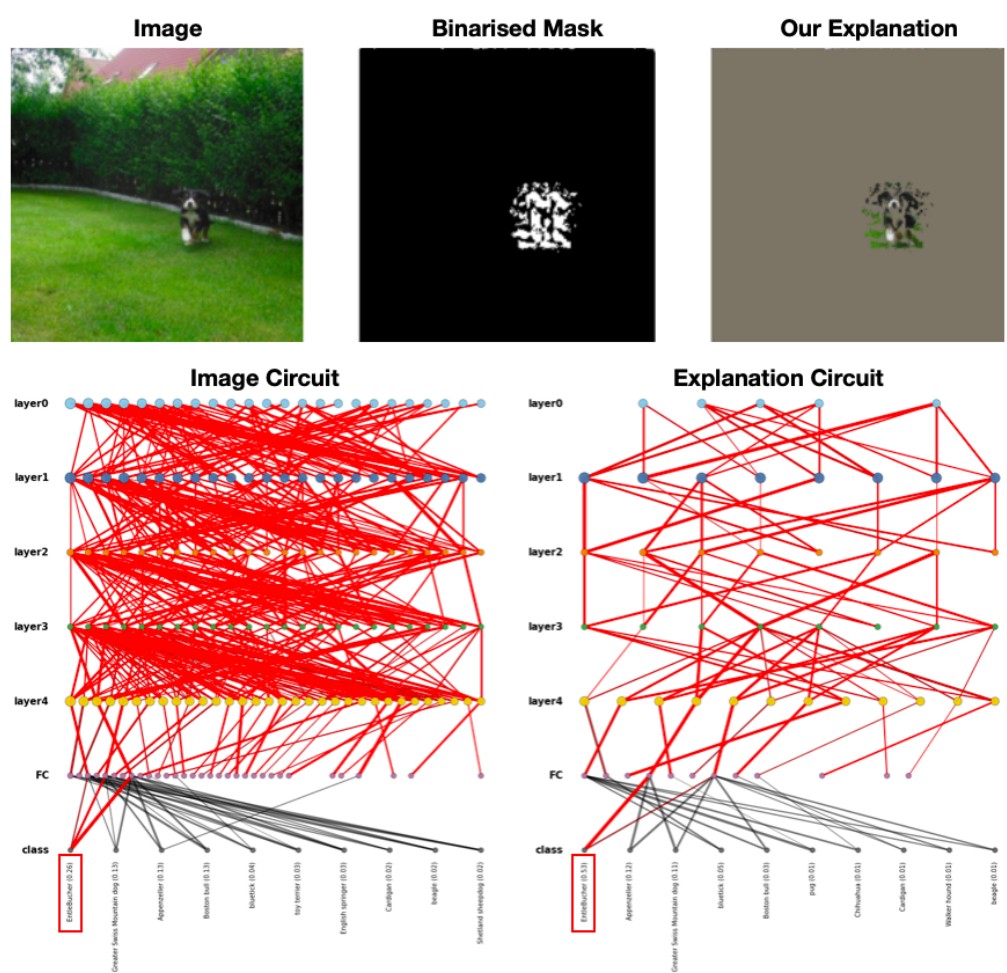

Figure 2: Top: Original Image, 0/1 Mask, and Explanation. Bottom: channel-level circuits derived from the original image and the explanation.

We observe that the explanation is highly minimal(only about 5% of active pixels), ignoring background regions of varying colors and textures, and focusing mostly on the object pixels. Also the confidence associated with the explanation goes up to 0.53 compared to that of 0.26 for the actual image as irrelevant background pixels have been turned off. Meanwhile the explanation circuit highlights only the dominant pathways necessary for the decision.

## 5 ABLATIONS

In Figure 3, we systematically examine how the inclusion and relative weighting of different loss terms impacts the resulting explanations. When using only the activation matching losses, the explanation degenerates to nearly the entire input image, since there is no explicit pressure to enforce sparsity. Also the use of total variation loss is necessary for generating noise free explanations. We therefore focus on the role of the area and total variation terms, which directly regulate the size and smoothness of the explanation masks. In the first case, heavily weighting both losses produces a compact mask that isolates only a small discriminative region, demonstrating the ability of our method to extract minimal yet sufficient evidence. In the second example, the explanation reveals a case of shortcut learning: the mask highlights not only the dog but also the leash, reflecting biases encoded in the training data. In the third example, relaxing the minimality constraints leads to broader masks that cover the dog more completely. Finally, when the area loss is excluded, the explanation expands to cover the full object, resembling a segmentation mask.

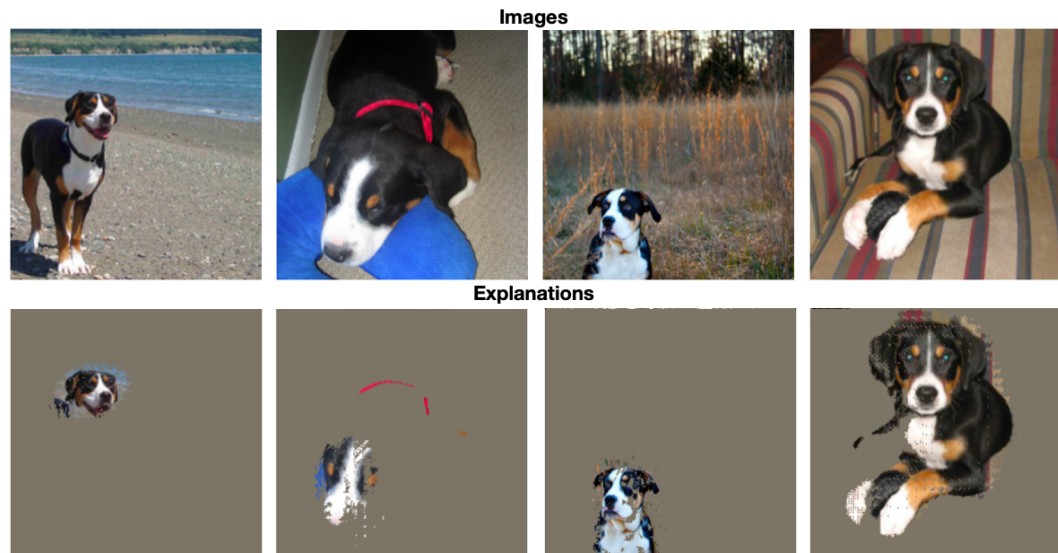

Figure 3: Effect of varying loss weights on generated explanations.

## 6 COMPARISONS

To contextualize the strengths and limitations of our method, we compare it against several widely used explanation techniques highlighting both qualitative difference is highlighting the relevant regions in the image and quantitative differences in terms of minimality. Specifically, we examine three families of baselines: (i) gradient-based attribution methods such as Grad-CAM, which visualize class-specific saliency through backpropagation; (ii) attention maps from transformer-based models, which expose self-attention patterns as proxies for importance; and (iii) abductive explanation approaches.

### 6.1 COMPARISONS WITH GRAD-CAM

We begin by comparing our explanations against Grad-CAM visualizations across a variety of models and input images in the left panel of Figure 4. The first example is an image classified as *house finch* by EfficientNet with a confidence of 0.11. Using our method with loss weights $\lambda_{\text{act}} = 1.0$, $\lambda_{\text{CE}} = 4.0$, $\lambda_{\text{KL}} = 0.4$, $\lambda_{\text{area}} = 15.0$, $\lambda_{\text{bin}} = 0.3$, $\lambda_{\text{tv}} = 15.0$, and $\lambda_{\text{rob}} = 6.0$, we obtain an explanation that sharply highlights the bird itself. In contrast, Grad-CAM focuses on the beak and background, producing a more diffused attribution.

The next example is an image of a *zucchini* classified by EfficientNet with confidence 0.59. Our explanation, generated with $\lambda_{\text{act}} = 2.0$, $\lambda_{\text{CE}} = 16.0$, $\lambda_{\text{KL}} = 5.0$, $\lambda_{\text{area}} = 45.0$, $\lambda_{\text{bin}} = 3.0$, $\lambda_{\text{tv}} = 23.0$, and $\lambda_{\text{rob}} = 20.0$, yields a confidence of 0.89 and minimally highlights one of the zucchinis in the scene. Grad-CAM, by comparison, highlights a larger overlapping region that covers two zucchinis simultaneously. The third example is classified as *granny smith* apples, by ConvNeXt with a confidence of 0.10. Using ($\lambda_{\text{act}} = 0.8$, $\lambda_{\text{CE}} = 10.0$, $\lambda_{\text{KL}} = 0.64$, $\lambda_{\text{area}} = 24.0$, $\lambda_{\text{bin}} = 1.8$, $\lambda_{\text{tv}} = 15.0$, $\lambda_{\text{rob}} = 2.0$), the explanation achieves a confidence of 0.62 while focusing on the edge and central region. Grad-CAM, however, spreads attention across multiple apple boundaries with substantial background included.

The next image of *jackfruit*, is classified by MobileNet with confidence 0.62. Our explanation, generated using $\lambda_{\text{act}} = 0.5$, $\lambda_{\text{CE}} = 5.0$, $\lambda_{\text{KL}} = 0.54$, $\lambda_{\text{area}} = 5.3$, $\lambda_{\text{bin}} = 1.4$, $\lambda_{\text{tv}} = 2.7$, $\lambda_{\text{rob}} = 3.2$, highlights minimal texture characteristic of jackfruit, raising the confidence to 0.92. Grad-CAM, in contrast, erroneously attributes saliency to wide regions overlapping across. Finally, we examine an image classified as *window shade* by EfficientNet with confidence 0.22. Using $\lambda_{\text{act}} = 1.5$, $\lambda_{\text{CE}} = 25.0$, $\lambda_{\text{KL}} = 7.5$, $\lambda_{\text{area}} = 80.0$, $\lambda_{\text{bin}} = 2.5$, $\lambda_{\text{tv}} = 35.0$, $\lambda_{\text{rob}} = 25.0$, we highlight the window

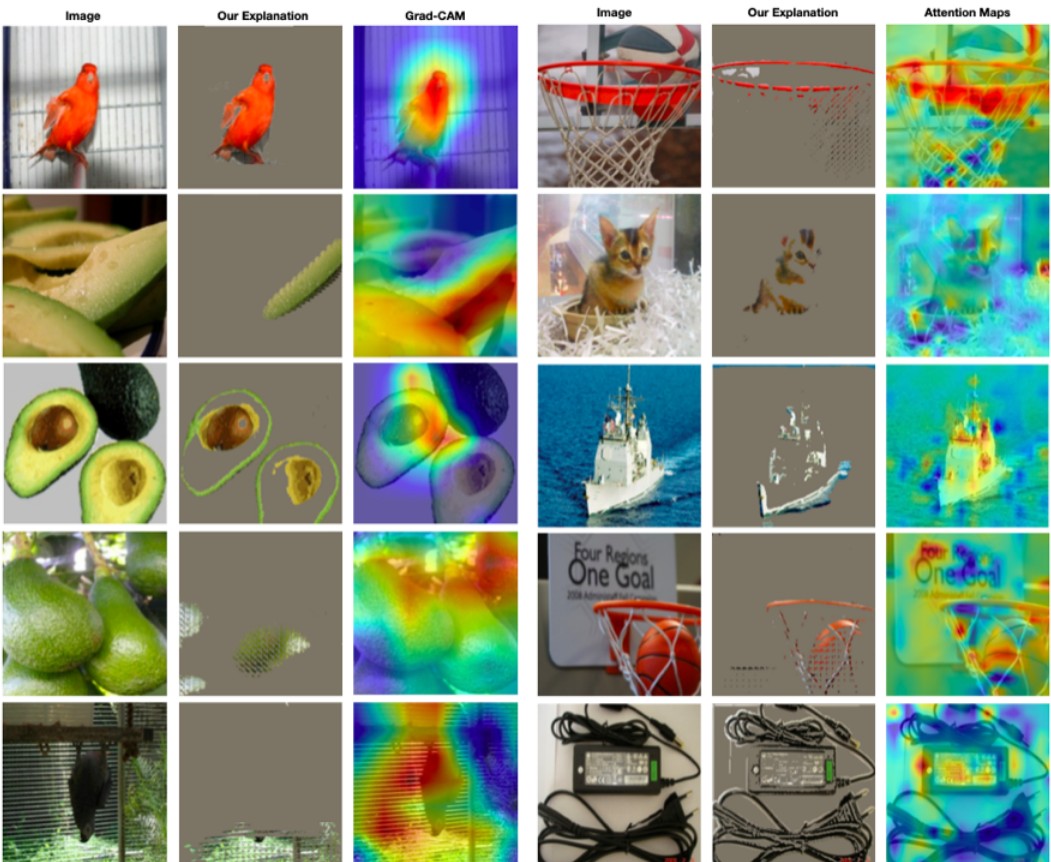

Figure 4: Left: Comparison with Grad-CAM. Right: Comparison with Attention Maps.

shade at the bottom of the image with 0.78 confidence, while discarding the bird in the foreground. Grad-CAM, however, allocates saliency to both the bird and the shade, diluting the explanation.

Together, these comparisons demonstrate that while Grad-CAM often highlights broad, overlapping regions with background leakage, our approach consistently produces sharper, minimal, and decisional-equivalent explanations that more faithfully capture the evidence underlying each classification.

## 6.2 COMPARISONS WITH ATTENTION MAPS

We next compare our explanations with attention maps extracted from ViT-16 shown on the right of Figure 4. The first example is an image classified as *basketball* with confidence 0.89. Using our loss weights $\lambda_{\text{act}} = 4.5$, $\lambda_{\text{CE}} = 15.0$, $\lambda_{\text{KL}} = 5.0$, $\lambda_{\text{area}} = 55.0$, $\lambda_{\text{bin}} = 1.5$, $\lambda_{\text{tv}} = 57.0$, and $\lambda_{\text{rob}} = 45.0$, the explanation minimally highlights only the ring that determines the classification, yielding a confidence of 0.90. Attention maps, however, emphasize broader regions dominated by red color patches, diluting the evidence.

The second example is an image of an *Egyptian cat*, classified with 0.41 confidence. Our explanation, generated with $\lambda_{\text{act}} = 4.5$, $\lambda_{\text{CE}} = 15.0$, $\lambda_{\text{KL}} = 5.0$, $\lambda_{\text{area}} = 56.0$, $\lambda_{\text{bin}} = 1.5$, $\lambda_{\text{tv}} = 28.0$, $\lambda_{\text{rob}} = 28.0$, is highly minimal, focusing on small, distinct regions of the cat. In contrast, attention maps spread widely across the entire image, with much weaker localization.

The next is an *aircraft carrier* classified with 0.76 confidence. Our explanation discards background waves entirely and concentrates only on the ship, raising the classification confidence to 0.81. Attention maps, in comparison, significantly highlights some regions of the carrier and also the swaths of the sea, making the attribution less precise. Another example with a *basketball* image further

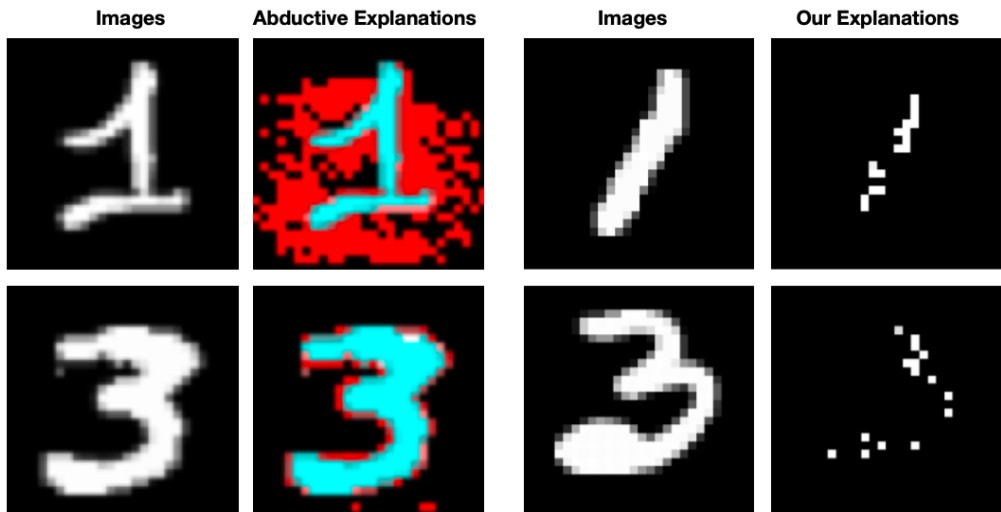

Figure 5: Left: Abductive Explanations. Right: Our Explanations

illustrates this contrast. Attention maps simultaneously focus on background text and the net, while our method, using $\lambda_{\mathrm{act}} = 4.5$, $\lambda_{\mathrm{CE}} = 15.0$, $\lambda_{\mathrm{KL}} = 5.0$, $\lambda_{\mathrm{area}} = 55.0$, $\lambda_{\mathrm{bin}} = 1.5$, $\lambda_{\mathrm{tv}} = 27.0$, $\lambda_{\mathrm{rob}} = 28.0$, exclusively highlights the basket and the ball itself.

Finally, in an image where a *charger* is misclassified as a radio with confidence 0.28, our explanation isolates the true object by excluding the background, raising the confidence to 0.91. Attention maps, however, primarily emphasize the adapter box, again highlighting irrelevant cues. Overall, these examples show that attention maps often diffuse across color regions or background context, while our approach produces compact, minimal, and class-faithful explanations that better align with human intuition.

### 6.3 COMPARISONS WITH ABDUCTIVE EXPLANATIONS

We also compare against abductive explanations using a custom three-layer CNN trained on MNIST digits as shown in Figure 5). In this setup, the abductive explanations highlight in red and blue the minimal evidence required for the prediction, shown on the left of each example. Our method, shown on the right, instead identifies the minimal set of pixels that not only preserves the predicted class label but also maintains the classifier's confidence relative to the original input.

It can be observed that abductive explanations typically include large contiguous regions of the digit as sufficient evidence for prediction. By contrast, our explanations generated with loss weights $\lambda_{\mathrm{act}} = 0.6$, $\lambda_{\mathrm{CE}} = 4.0$, $\lambda_{\mathrm{KL}} = 0.54$, $\lambda_{\mathrm{area}} = 100.0$, $\lambda_{\mathrm{bin}} = 1.2$, $\lambda_{\mathrm{tv}} = 50.0$, and $\lambda_{\mathrm{rob}} = 10.0$ are far more compact: across all MNIST classes, only about 1–2% of the pixels remain active, while still preserving both the label and the confidence. This demonstrates that our approach yields sharper, more faithful explanations that capture the truly decisive strokes of each digit, rather than broad swaths of input space.

## 7 CONCLUSION

We presented an activation-matching–based framework for generating minimal and faithful explanations of pre-trained image classifiers using an autoencoder that learns to produce binary masks discarding irrelevant pixels in the explanations. Beyond input-level interpretability, we further showed how these explanations can be coupled with forward activations and backward gradients to uncover sparse computational sub-circuits that realize individual decisions within deep networks. As future work, a natural direction is to explore formal guarantees of minimality in the generated explanations, strengthening the theoretical foundation of our approach while extending its applicability to broader domains and architectures.

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

# A    MORE RESULTS

As shown in Figure 6, when strong minimality constraints are applied, the explanation for a single otter reduces to a remarkably small region—roughly 2% of pixels—focusing primarily on the facial features and fur texture. Despite this extreme sparsity, the classifier's label is preserved with high confidence. In contrast, when applied to an image with multiple otters, the method produces separate explanations that selectively attend to each animal, demonstrating how the approach can adapt to multi-instance settings and highlight distinct decision-supporting evidence for each occurrence.

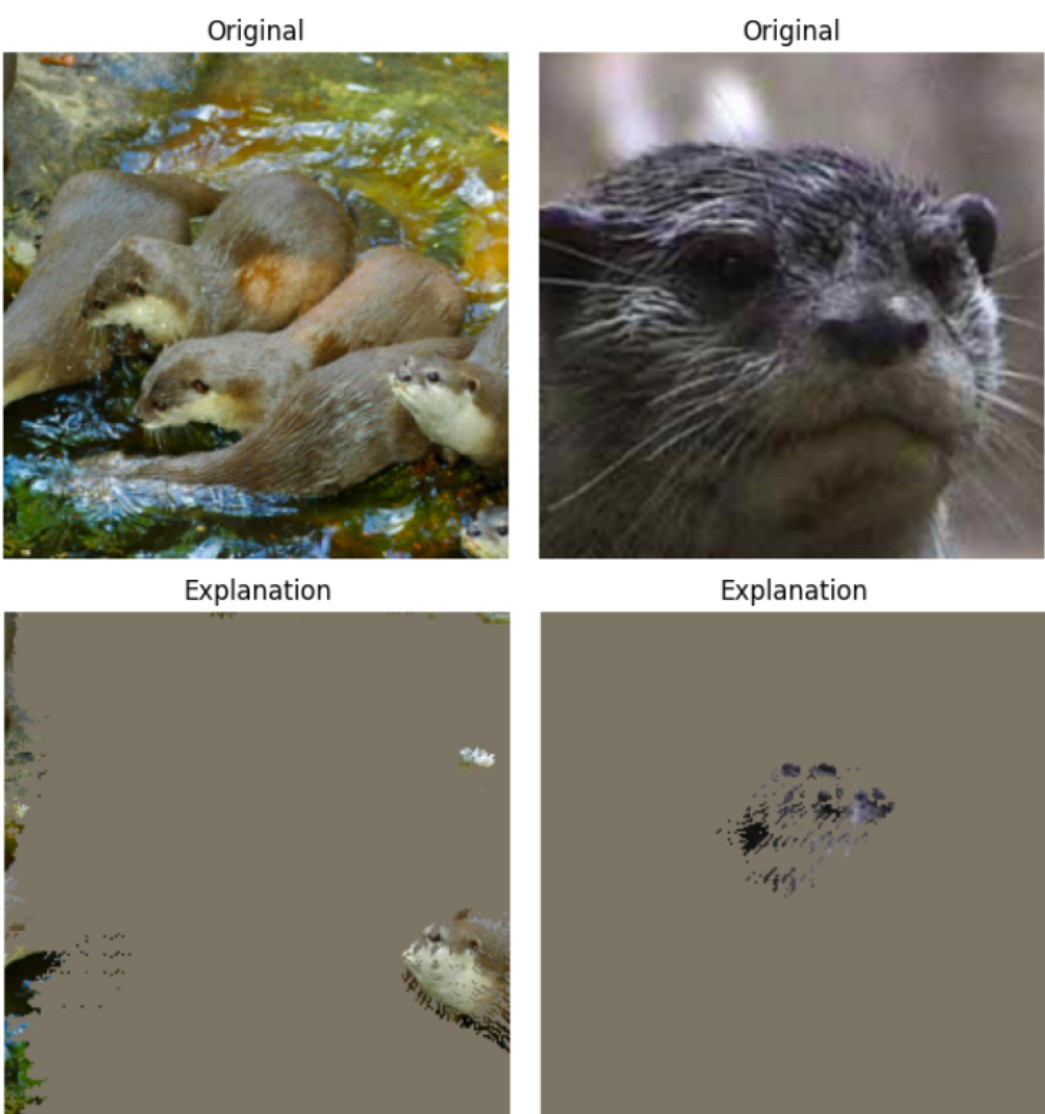

Figure 6: Explanations for sample Images of *Otter*

Figure 7 illustrates how our method sheds light on model errors. In the first example, an image of a wooden toilet seat is misclassified as a paint brush. The generated explanation reveals that the model focused almost entirely on the decorative print on the seat rather than the seat's structure, explaining the spurious prediction. In another case, an image of a stinkhorn mushroom is misclassified as a goldfish. Here, the explanation highlights background regions alongwith the actual fungus, showing that the model mostly ignored the object of interest and instead relied on irrelevant context.

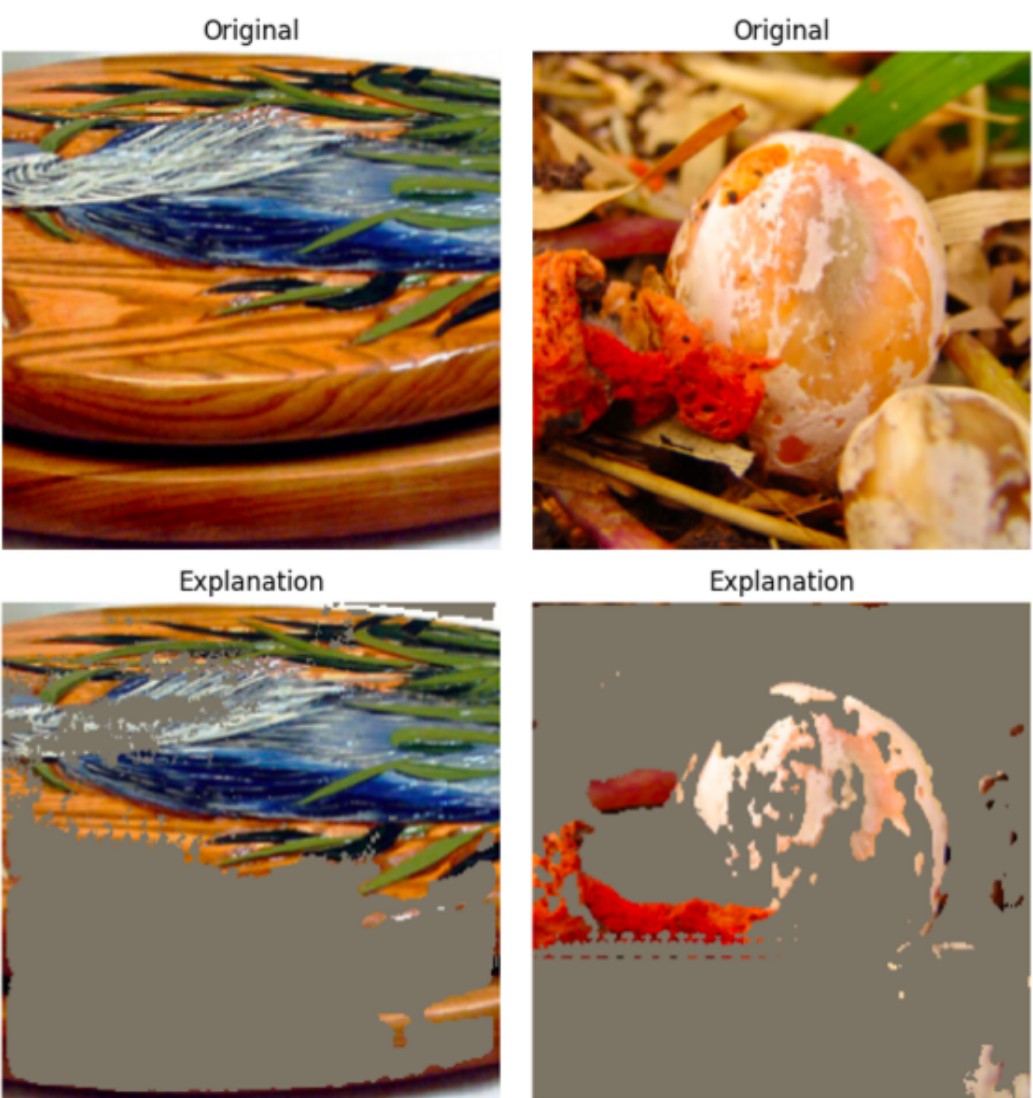

Figure 7: Explanations for misclassified Images.

We further evaluate our framework on MNIST digits using the same custom three-layer CNN and the same set of loss weights as in the abductive comparison ($\lambda_{\text{act}} = 0.6$, $\lambda_{\text{CE}} = 4.0$, $\lambda_{\text{KL}} = 0.54$, $\lambda_{\text{area}} = 100.0$, $\lambda_{\text{bin}} = 1.2$, $\lambda_{\text{tv}} = 50.0$, $\lambda_{\text{rob}} = 10.0$). Figure 8 shows a row of original digit images followed by the corresponding explanations generated by our method. In each case, the autoencoder produces masks that preserve both the predicted class label and its confidence, while activating only about 1–2% of the pixels. This demonstrates that even for simple architectures, our approach isolates extremely sparse yet sufficient evidence, capturing only the decisive strokes of each digit without relying on broader context. The resulting explanations are not only faithful to the classifier's decision but also concise and visually intuitive.

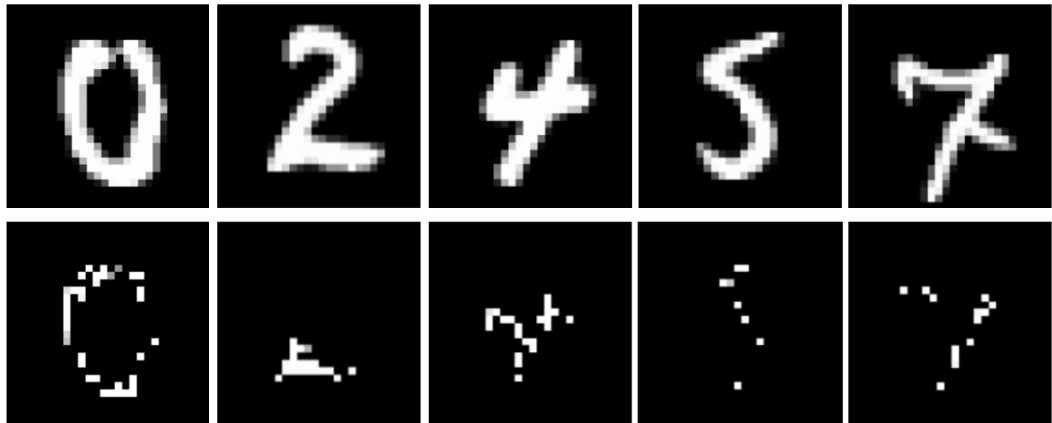

Figure 8: Explanations for MNIST digits.

