# OpenReview forum: "Minimalist Explanation Generation and Circuit Discovery"
_ICLR.cc/2026/Conference — ICLR 2026 Conference Withdrawn Submission_

### Official Review · Reviewer_btnU · 2025-10-18

**Soundness:** 1
**Presentation:** 3
**Contribution:** 1
**Rating:** 0
**Confidence:** 5

**Summary:**

The paper aims to identify minimal explanations that not only preserve the model’s decision but are also concise and human-readable. A lightweight autoencoder is trained to produce binary masks that highlight decision-critical regions of an image while discarding irrelevant background. Additionally, the work introduces a circuit readout procedure, where, by using the explanation’s forward pass and gradients, active channels are identified, and a channel-level graph is constructed.

**Strengths:**

1. Clarity: The paper is overall written clearly.
2. Beyond Input-Level Explanations: The approach provides insights into the network’s internal computations, offering a window into its decision-making process. However, it is still not entirely clear how to understand the explanation circuit from a user perspective, as shown in Figure 2.

**Weaknesses:**

1. Hyperparameter Balancing: The training objective contains multiple terms, but the method for deciding the hyperparameters to balance these terms is unclear. In Section 6.1, different hyperparameters are used for different explanations, but no justification or systematic approach is provided.


2. Lack of Quantitative Results: The paper does not include any quantitative experimental results. Only a few visual examples are provided, comparing the method to Grad-CAM and attention maps.

3. Understanding the Explanation Circuit: While the circuit readout procedure is introduced, it remains unclear how users can interpret and learn information from the explanation circuit.

4. Computational Cost: The explanation method is computationally expensive, especially when compared to white-box explanation methods such as Grad-CAM.

**Questions:**

See weaknesses above.

---

### Official Review · Reviewer_LS6g · 2025-10-29

**Soundness:** 1
**Presentation:** 2
**Contribution:** 2
**Rating:** 2
**Confidence:** 3

**Summary:**

This paper proposes a framework for generating classification explanations and identifying subnetworks that support such predictions in image models.  Explanations are produced by applying a learned binary mask to the input image, where the mask is generated by a U-Net trained to minimize a multi-objective loss computed from the image model’s activity and aspects of the explanation mask.  The resulting explanation image preserves the model’s decision while highlighting the relevant regions for its prediction.  Circuit discovery is performed by ranking channels in the frozen network based on activation energy, weighted by connection weights and gradients. The authors conduct a qualitative ablation study on loss components and compare their explanations with grad-cam, attention maps, and abductive explanations.

**Strengths:**

- The proposed framework is general to image models and intuitive.  It places no assumptions on the model architecture itself, permitting broad usability.
- The paper is well motivated: explaining model behavior and identifying important subcircuits remains an important research question.
- Resulting explanations and subcircuits shown in paper preserve model output behavior.  This is accomplished with far fewer pixels and channels than used in the original, frozen network.

**Weaknesses:**

- Results and qualitative comparisons to alternative methods for output explanation are subjective and unconvincing.
- No intuition is provided about how to select the many hyperparameters of the model and how to address tradeoffs between mask sparsity and frozen model activity matching.
- Ablation study is qualitative and does not present any clear criteria for what makes one explanation better than another.  Conflicting ideas are presented: for example, in Figure 3,  the authors suggest the importance of the area loss to prevent learning full-coverage object masks but in the explanation for the parrot in Figure 4, they suggest that this explanation (which covers the full object) is indeed high quality.
- No quantitative results or analyses are provided for minimality of explanations or circuits.  Is it true that these are empirically minimal, and cannot be further reduced with more masking or circuit pruning?

**Questions:**

- Can you provide any evidence of minimality for explanations or circuits?  For instance, removing any additional pixels or connections make the model predictions empirically worse?
- Are there any ways that you can more broadly quantify the utility of explanations, and quantitatively compare these results to other methods such as grad-cam, abductive ablations, and attention maps? Can you quantify (over an entire validation set) minimality, output consistency, etc. in the proposed network and comparable baselines (can consider generating masks by simply thresholding explanation maps of baselines)?
- Many hyperparameters are involved and there is not clear criteria for how to optimize these.  Further, in results section 6.2, hyperparameters differ for differing images despite holding the network constant as EfficientNet or MobileNet.
- How is the U-Net trained?  It is trained over an entire training dataset?

---

### Official Review · Reviewer_wmfU · 2025-10-31

**Soundness:** 3
**Presentation:** 2
**Contribution:** 2
**Rating:** 2
**Confidence:** 4

**Summary:**

This paper proposes a framework towards minimal and faithful explanations for image encoders aiming to bridge some gaps in input-level and mechanistic interpretability. The key idea is to produce sparse binary masks that highlight the smallest set of pixels sufficient to preserve a model’s decision while maintaining internal activation similarity. This idea of combining activation-matching explanation masks over the input and circuit discvery is appealing, while the proposed unified framework is interesting.

**Strengths:**

The paper introduces an interesting combination of input-level explanation and mechanistic interpretability, bridging two strands of research that are typically addressed separately and the method is model agnostic.

I find the use of binary masks much more important that other methods like GradCAM and attention maps since it yields more interpretable regions with visually compact information,.

**Weaknesses:**

My main concerns focus on the experimental setting and presentation of the results. Specifically:

*Loss design and scaling*: The framework uses many different losses, each with separate scales and empirically tuned weights. Different examples use distinct $\lambda$-weight configurations, making it unclear how robust the method is or how much of the visual outcome depends on tuning. Can the authors justify the necessity and relative scaling of each loss term? I would also like to see an ablation on the effect of the parameters on some examples.

*KL divergence* term: The inclusion of KL divergence to align predictive distributions is only briefly motivated without an insightful justification. Why is this loss required in addition to cross-entropy? Does it meaningfully affect explanation quality or training stability?

*Activation matching choice*: The method aligns post-ReLU activations rather than pre-ReLU ones, which removes negative activations and may affect representational fidelity. Why was this design choice made? What happens if activations are matched pre-ReLU instead?

*Mask Minimality*: The auhors mention throughout the text "minimal" explanations. The use of a mask sparsity constraint does not guarantee theoretical minimality, just compactness in the representation. Is there any evidence or derivation to back this claim?

*Circuit Discovery*: The “circuit discovery” stage is essentially an activation summary, selecting top-k channels by activation energy and overlaying gradients for attribution. While this visualization is interesting, it does not necessarily imply the discovery of minimal or causal subcircuits. The masking process itself does not guarantee a minimal representation (p.5 l. 246).

*Experimental setup and comparisons*:
- The first set of experiments do not provide in my view any substantial information about circuit discovery. The images are low quality, while only one example is examines.
- Ablations are shown but with no quantitative results or performnace metrics (again on a single example).
- Comparisons with Grad-CAM and attention maps rely on handpicked examples with differing weights, which raises serious questions about the tunability of the proposed framework.
- Some results (e.g., “avocados 2,” “parrot,” “watermelon” in Figure 4) are difficult to interpret or seem inconsistent with the described behavior.

*Architecture Choice*: The explanation generator is a U-Net encoder pretrained for mask generation, rather than being trained end-to-end from scratch. Does this bias the explanation outcomes? Why not train the encoder directly on the task?

**Questions:**

Please see the Weaknesses section.

---

### Official Review · Reviewer_qThL · 2025-10-31

**Soundness:** 3
**Presentation:** 3
**Contribution:** 2
**Rating:** 2
**Confidence:** 4

**Summary:**

The paper introduces a learnable explanation method that aims for human interpretability of computer vision neural network models using penalties to keep them minimal. As the explanations are given in the form of masks and circuits over the neural network, "minimal" here means two things: explanation masks over the input image should cover a small area of the image while being as contiguous as possible; circuits along the network should be not dense. The method is then evaluated qualitatively against known methods employed for explainability.

**Strengths:**

The paper is very well written, with a smooth flow between the sections, sound motivation, contextualisation among existing methods, and clear method description. The method is also simple to understand and implement, which is a bonus.

Being quite intuitive, the proposed method follows a reasonable set of principles to guide the design of the objective function being optimised. Furthermore, careful design decisions such as keeping explanations simple to understand are also appreciated.

**Weaknesses:**

While intuitive, the method relies heavily on hyperparameters, which is due to its ad-hoc nature. This is a problem I see with this type of approach: the lack of theory for specifying the problem leads to the creation of another black box (autoencoder + non-trivially related loss functions) to explain the first.

With this in mind, I wonder what it is that the method brings to the table (Question 1 below) in terms of advancing the subfield of explainability methods. Experimentally, only qualitative evaluations are present, which can look convincing, but does not constitute exhaustive evidence for the method, in my view. Visually, I see minor novelty compared to existing methods (what can be concluded from explanations computed with different methods remains the same).

**Questions:**

1. What do the generated explanations make visible that was not possible before?
2. How does one interpret a circuit? It seems barely possible in larger models, where interpretability is more crucial. Can you clarify this?
3. Why is there no comparison to LRP (Bach et al., 2015)? The method seems to be also relevant and highly cited.
4. Is a badly trained mask generator easily diagnosed? In that same vein, does that mean one can quickly discard not-so-good mask generators and avoid interpretability pitfalls?
5. In lines 413-414, the authors write their produce "explanations that more faithfully capture the evidence underlying each classification." What does an explanation from the proposed method look like in a case where the classifier uses the background to reach a decision?

**Other comments:**

- l. 160: used to preserves => preserve

**References:**

Bach, S., Binder, A., Montavon, G., Klauschen, F., Müller, K. R., & Samek, W. (2015). On pixel-wise explanations for non-linear classifier decisions by layer-wise relevance propagation. PloS one, 10(7), e0130140.

---

### Note · Authors · 2025-11-23

I have read and agree with the venue's withdrawal policy on behalf of myself and my co-authors.